# Low irradiance multiphoton imaging with alloyed lanthanide nanocrystals

Bining Tian[1], Angel Fernandez-Bravo[1], Hossein Najafiaghdam[2], Nicole A. Torquato[1,3], M. Virginia P. Altoe[1], Ayelet Teitelboim[1], Cheryl A. Tajon[1], Yue Tian[1], Nicholas J. Borys[1], Edward S. Barnard[1], Mekhail Anwar[3], Emory M. Chan[1], P. James Schuck[1,4] & Bruce E. Cohen[1]

Multiphoton imaging techniques that convert low-energy excitation to higher energy emission are widely used to improve signal over background, reduce scatter, and limit photodamage. Lanthanide-doped upconverting nanoparticles (UCNPs) are among the most efficient multiphoton probes, but even UCNPs with optimized lanthanide dopant levels require laser intensities that may be problematic. Here, we develop protein-sized, alloyed UCNPs (aUCNPs) that can be imaged individually at laser intensities >300-fold lower than needed for comparably sized doped UCNPs. Using single UCNP characterization and kinetic modeling, we find that addition of inert shells changes optimal lanthanide content from $Yb^{3+}$, $Er^{3+}$-doped $NaYF_4$ nanocrystals to fully alloyed compositions. At high levels, emitter $Er^{3+}$ ions can adopt a second role to enhance aUCNP absorption cross-section by desaturating sensitizer $Yb^{3+}$ or by absorbing photons directly. Core/shell aUCNPs 12 nm in total diameter can be imaged through deep tissue in live mice using a laser intensity of $0.1\,W\,cm^{-2}$.

[1] The Molecular Foundry, Lawrence Berkeley National Laboratory, Berkeley, CA 94720, USA. [2] Department of Electrical Engineering and Computer Sciences, University of California, Berkeley, Berkeley, CA 94720, USA. [3] Department of Radiation Oncology, University of California, San Francisco, San Francisco, CA 94158, USA. [4] Department of Mechanical Engineering, Columbia University, New York, NY 10027, USA. Correspondence and requests for materials should be addressed to E.M.C. (email: emchan@lbl.gov) or to P.J.S. (email: p.j.schuck@columbia.edu) or to B.E.C. (email: becohen@lbl.gov)

Light microscopy is the primary means of studying complex living systems, enabling real-time analysis with ever-increasing spatial and temporal resolution. Increasingly powerful imaging techniques and lasers have raised concern over light toxicity[1,2], which is most acute with high laser intensities at shorter wavelengths in the ultraviolet and visible regions[3,4]. Near-infrared (NIR) excitation is more benign than these higher energy wavelengths[4,5], and nonlinear multiphoton techniques that use NIR excitation have been widely adopted[6–10]. Both scatter and absorption by cellular components are much lower for NIR light than for visible light[8,11,12], and this steep wavelength dependence has been shown in direct comparisons to reduce photodamage using NIR-based techniques[4,5,13,14]. Multiphoton probes excitable at reduced laser intensities in the NIR would enable powerful high-resolution and deep-tissue imaging techniques in sensitive systems without associated phototoxicity.

Lanthanide-doped upconverting nanoparticles (UCNPs) are phosphors that absorb multiple photons in the NIR and emit at higher energies in the NIR or visible spectral regions. The luminescence efficiencies of UCNPs are orders of magnitude higher than those of the best two-photon fluorophores[15–18], and they exhibit no on-off blinking, no overlap with cellular auto-fluorescence, and no measurable photobleaching under prolonged single-particle excitation[5,19,20]. UCNPs make use of energy transfer upconversion between neighboring lanthanide ions ($Ln^{3+}$), in which sensitizer ions with relatively large absorption cross-sections sequentially transfer absorbed energy to luminescent emitter ions, both of which are doped into a low-phonon-energy nanocrystal host. For many applications, $\beta$-phase $NaYF_4$ nanocrystals doped with 20% $Yb^{3+}$ sensitizer and a low percentage of $Er^{3+}$ or $Tm^{3+}$ emitter are most efficient[17,18,21]. Addition of inert epitaxial shells to these UCNPs has been shown to enhance emission at low excitation powers by reducing $Yb^{3+}$-mediated energy migration to high-vibrational-frequency modes of surface oleate ligands or solvent[19]. For UCNPs with high $Ln^{3+}$ content, this has been attributed to suppression of concentration quenching[22,23], an observation that encompasses a number of known as well as unexplored energetic pathways that reduce the quantum yield (QY) of upconverted emission[22,24–28].

Here, we use single-nanoparticle characterization and kinetic models of $Ln^{3+}$ energy transfer to develop antibody-sized (approximately 12–15 nm diameter) alloyed UCNPs that can be imaged at the single-particle level at laser intensities below 300 W $cm^{-2}$, over 300-fold lower than needed for comparably sized doped UCNPs. Core/shell aUCNPs are brighter than comparably sized doped UCNPs at all laser intensities tested, over a range of four orders of magnitude. Addition of inert epitaxial shells radically changes optimal lanthanide content from $Yb^{3+}$, $Er^{3+}$-doped $NaYF_4$ nanocrystals to fully alloyed compositions, and at high levels of the emitter $Er^{3+}$, these ions can adopt a second role to enhance the effective aUCNP absorption. This leads to a revised UCNP design in which there is no need to dope $Ln^{3+}$ ions into an inert $NaYF_4$ (or other) host matrix. In live mice, aqueous 12-nm core/shell aUCNPs can be imaged with strong contrast (signal:background >25) through several millimeters of tissue with a laser intensity of just 0.1 W $cm^{-2}$. aUCNPs open up the possibility of using both low irradiance and low-energy excitation wavelengths for non-destructive bioimaging experiments.

## Results

### Characterization of small $NaLnF_4$ core/shell nanoparticles. To better understand how inert epitaxial shells affect UCNP emission, we analyzed emissions of single nanocrystals to compare absolute brightness at different laser intensities. UCNP emission is deeply power-dependent and size-dependent, and single-

nanocrystal characterization allows quantitative comparison of non-aggregated nanocrystals under identical environments over four orders of magnitude excitation power density[8,13], a range that spans imaging experiments from single-molecule studies to imaging of highly light-sensitive samples. We synthesized a series of 8-nm diameter $\beta$-phase $NaYF_4$ cores[19], and overcoated them with $NaYF_4$ shells using a layer-by-layer protocol[29] (Fig. 1 and Methods). Several $NaLnF_4$ alloys of heavy lanthanides (e.g., $Yb^{3+}$-$Er^{3+}$, $Yb^{3+}$-$Tm^{3+}$, and $Yb^{3+}$-$Ho^{3+}$, as well as $NaErF_4$) have been reported[22,25,28,30,31], including sub-20-nm $NaYbF_4$:Tm core/shell nanoparticles[32], although none of these compositions have not been characterized by quantitative single-particle imaging or systematically over a large range of power densities. Characterization of our nanocrystals by electron microscopy (EM) and X-ray diffraction (XRD) showed monodisperse $\beta$-phase nanocrystals for both 8-nm core and core/shell UCNPs (Fig. 1e and Supplementary Figs. 1–3). High-angle annular dark-field scanning transmission EM (STEM), which is sensitive to atomic number $Z$, confirms the core/shell structure, showing clear boundaries between alloyed cores and 20% $Gd^{3+}$-doped shells (Fig. 1f).

Upconverted emission spectra of single core/shell UCNPs have similar transitions as seen for unshelled UCNPs (Fig. 1d and Supplementary Fig. 4), but laser power density studies show large increases in emission dependent on both the lanthanide content and laser power densities. At laser intensities above $10^6$ W $cm^{-2}$, core UCNPs doped with 20% $Yb^{3+}$ sensitizer show maximum emission with 20% $Er^{3+}$ emitter (Fig. 2a), while higher $Er^{3+}$ doping leads to apparent $Er^{3+}$-$Er^{3+}$ cross-relaxation or enhanced surface losses[18,25]. Once shells are added, additional $Er^{3+}$ increases emission, with fully alloyed UCNPs almost an order of magnitude brighter (Fig. 2b) than the most efficient doped cores[18]. In contrast to unshelled UCNPs, where power dependence is critical and the brightest compositions at high laser intensities are not even luminescent at low intensities[18], here the brightest aUCNP compositions are superior over a 10,000-fold range of excitation intensities. At the highest excitation intensities, shells provide almost a twofold emission increase for 2% $Er^{3+}$ doping and increase up to 30-fold for aUCNPs (Fig. 2a and Supplementary Fig. 5), suggesting that at laser powers used for single-particle imaging, any quenching typically associated with high $Ln^{3+}$ content[25,26] can be suppressed by the addition of inert shells[22]. Single UCNP analysis of shell thickness (Fig. 2b and Supplementary Fig. 1) shows larger emission enhancements at lower powers, and with enhancements diminishing above 4-nm shell thickness[16]. Power series show little difference in single aUCNP brightness with either a majority of $Yb^{3+}$ sensitizer or $Er^{3+}$ emitter ions (e.g., $NaEr_{0.6}Yb_{0.4}F_4$ vs. $NaEr_{0.2}Yb_{0.8}F_4$; Supplementary Fig. 5), suggesting more complex behaviors than the traditional roles assigned to these ions.

### Imaging aUCNP nanoparticles at low irradiances. We observed an even greater effect of inert shells at the lowest intensities (Fig. 3a). While the best cores require $>10^5$ W $cm^{-2}$ to image single sub-20-nm nanocrystals at signal-to-background ratios (SBR) above 3 (corresponding here to 100 c.p.s. above background[18]), most individual core/shell UCNPs can be imaged at irradiances below 1000 W $cm^{-2}$ (Fig. 3b and Supplementary Fig. 6). We observe that increasing the fraction of $Yb^{3+}$ (Supplementary Fig. 5) enables single core/shell $NaEr_{0.2}Yb_{0.8}F_4$ aUCNPs to be imaged as low as 290 W $cm^{-2}$ (Fig. 4a–e and Supplementary Fig. 7), >300-fold lower than the best previous sub-20-nm compositions[17,18]. For confocal imaging of ensembles of nanoparticles, intensities as low as 4 W $cm^{-2}$ can be used to image these same aUCNPs loaded into polystyrene beads

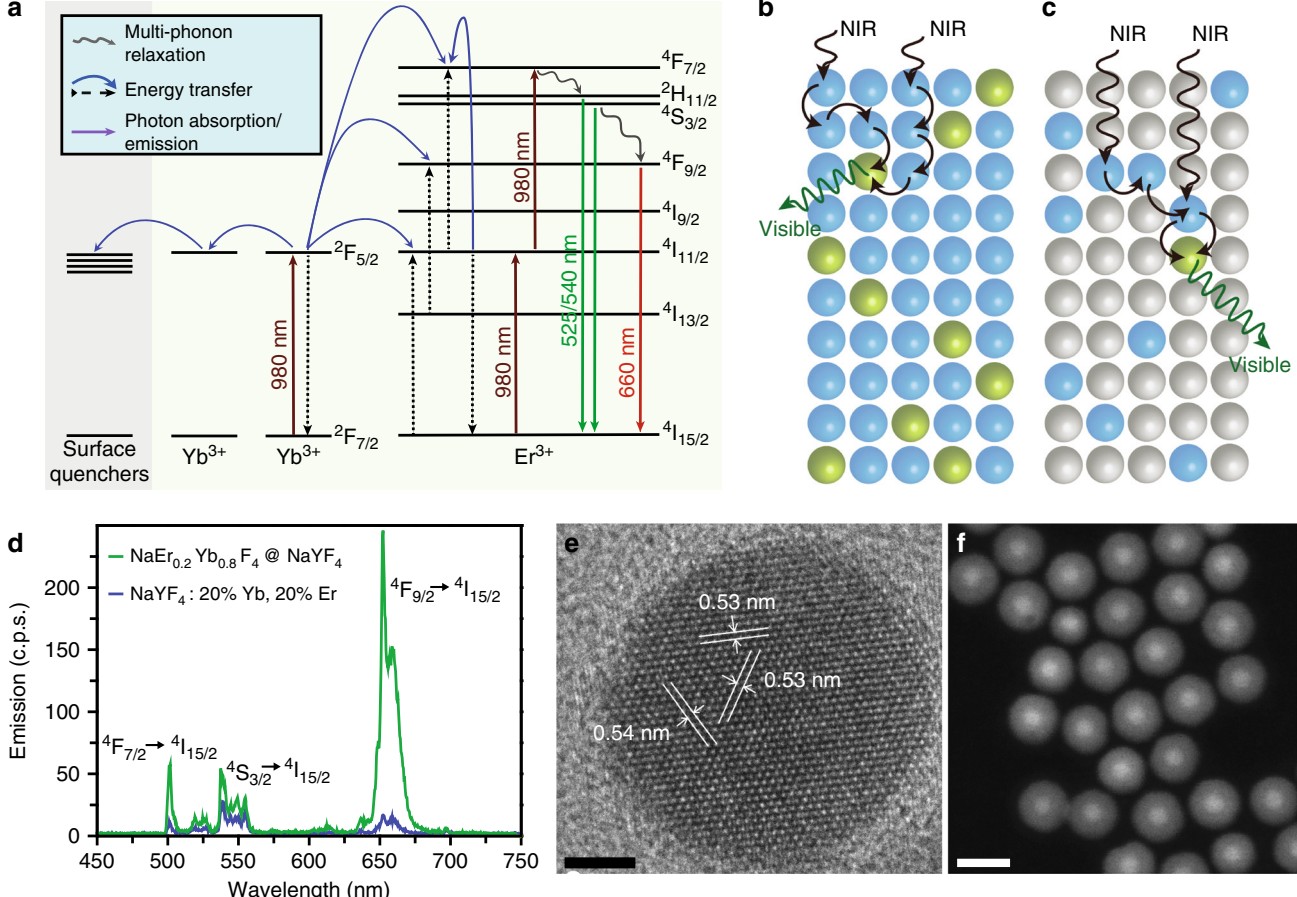

**Fig. 1** Nanoparticles for low irradiance imaging. **a** Multiphoton energy absorption, transfer, surface loss, and emission in $Yb^{3+}$/$Er^{3+}$ UCNPs and aUCNPs. Energy transfer pathways were calculated with a kinetic model as described[18, 33]. Comparison of alloyed (**b**) and doped (**c**) UCNPs, showing energy transfer in matrix $Y^{3+}$ (gray), sensitizer $Yb^{3+}$ (blue), and emitter $Er^{3+}$ (green) ions. **d** Emission spectra of single core/shell $NaYb_{0.8}Er_{0.2}F_4$ aUCNPs and $NaYF_4$: 20% $Yb^{3+}$, 20% $Er^{3+}$ UCNPs under $2 \times 10^6$ W $cm^{-2}$ 980 nm laser excitation. **e** High-resolution TEM image of an 8-nm $NaYb_{0.4}Er_{0.6}F_4$ with 4-nm $NaY_{0.8}Gd_{0.2}F_4$ shell (scale bar is 5 nm) and **f** high-angle annular dark-field STEM of 8-nm $NaYb_{0.4}Er_{0.6}F_4$ with 8-nm $NaY_{0.8}Gd_{0.2}F_4$ shells (scale bar is 25 nm)

(Fig. 4f–j and Supplementary Fig. 7), and intensities <0.1 W $cm^{-2}$ can be used for imaging aUCNPs injected into live mice, as discussed below. From these experiments, a revised UCNP design emerges in which there is no need to dope lanthanides into an inert matrix, as fully alloyed lanthanide UCNPs are brighter at all laser intensities.

**Mechanisms of improved aUCNP brightness**. To understand why the brightest core/shell compositions are alloyed, we determined single-nanocrystal absorption cross-sections and QYs (Supplementary Fig 8) and used these data with rate equation models[18,33] to uncover how high $Ln^{3+}$ content alters photophysical processes in UCNPs. At irradiances above the saturation intensity, $I_s$, at least 25% of $Yb^{3+}$ ions are populated in their $^2F_{5/2}$ excited state and thus are not able to absorb additional incident 980 nm photons. Calculations of $I_s$ based on $Yb^{3+}$ and $Er^{3+}$ absorption cross-sections at 980 nm ($\sigma_{abs}$; Supplementary Fig. 8a and Methods) show that core/shell aUCNPs reach saturation at 3000 W $cm^{-2}$, and kinetic models of UCNP $Ln^{3+}$ populations[13] find that ≥75% of $Yb^{3+}$ are in the excited state at intensities above $10^6$ W $cm^{-2}$. Although the calculated $\sigma_{abs}$ for $Yb^{3+}$ at 980 nm is 14-fold larger than that of $Er^{3+}$ (see Methods), at high laser intensities, the contribution of the remaining ground state $Yb^{3+}$ to total UCNP absorption should be sharply diminished, and even eclipsed by the $Er^{3+}$ contribution in $NaEr_{0.8}Yb_{0.2}F_4$

aUCNPs. However, with increasing $Er^{3+}$ levels, $Yb^{3+}$ ET to nearby $Er^{3+}$ ions is enhanced, as ET rates scale with $R^{-6}$ (where $R$ is the distance between ions) and with the product of the ion concentrations. This suggests that nearby $Er^{3+}$ ions offer new pathways to return $Yb^{3+}$ to its ground state, and this desaturation would then enhance the effective absorption cross-section of the aUCNP (Fig. 5).

**New roles of $Er^{3+}$ in enhanced aUCNP absorption**. To determine whether desaturation and direct $Er^{3+}$ absorption lead to the enhanced emission of aUCNPs, we determined single UCNP QYs based on single UCNP emissions (Fig. 2a) relative to a well-established doped UCNP standard[13] (Fig. 3c). UCNP absorption cross-sections were calculated (Supplementary Fig. 8) for the saturating intensities of the single UCNP measurements ($2 \times 10^6$ W $cm^{-2}$). For UCNPs with 20% $Yb^{3+}$, QY values rise sharply from 2 to 20% $Er^{3+}$, but above 20% QY values plateau while relative single UCNP emission continues to rise. This suggests relief from quenching is most important at lower $Er^{3+}$ content, while $Er^{3+}$ above 20% leads to increasing emission by enhanced absorption. Comparing the brightest aUCNPs to doped UCNPs with the same $Er^{3+}$ content, additional $Yb^{3+}$ leads to unchanged or decreased QY values (e.g., compare 20% $Yb^{3+}$, 60% $Er^{3+}$, and $NaEr_{0.6}Yb_{0.4}F_4$ in Fig. 3c). In contrast with previous discussions of high $Ln^{3+}$ UCNPs,

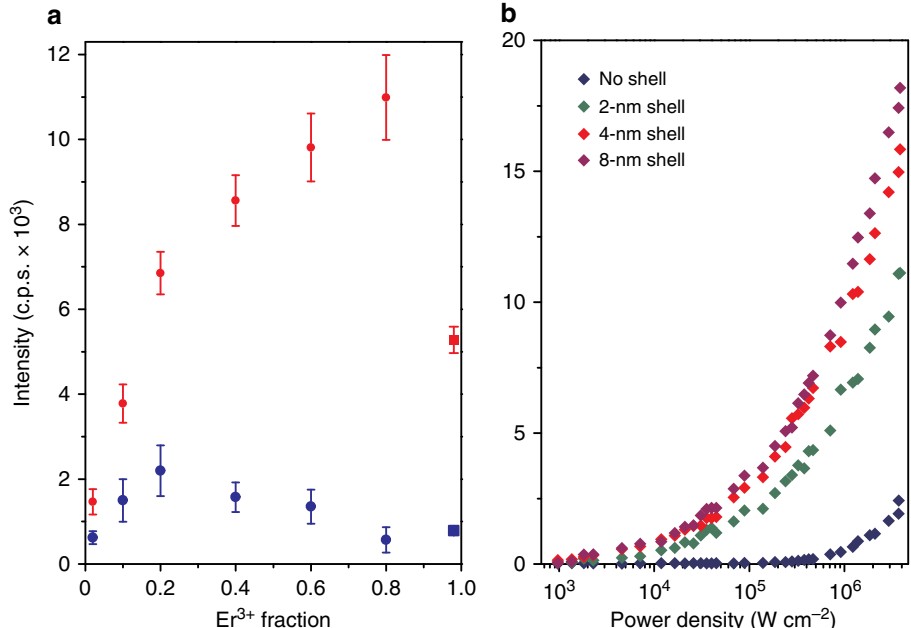

**Fig. 2** Enhancement of single UCNP emission. **a** Single-particle emission (496–745 nm) as a function of $Er^{3+}$ content. Blue and red circles are 8 nm $NaY_{(0.8-x)}Er_xYb_{0.2}F_4$ cores and with 4 nm shells, respectively. Blue and red squares are $NaErF_4$ and $NaErF_4$ core/shells, respectively. Excitation density is $2 \times 10^6$ W cm$^{-2}$. Error bars are one standard deviation from the mean ($n \geq 50$). **b** Laser intensity-dependent effects of inert shell thickness on single $NaYb_{0.4}Er_{0.6}F_4$ aUCNP emission

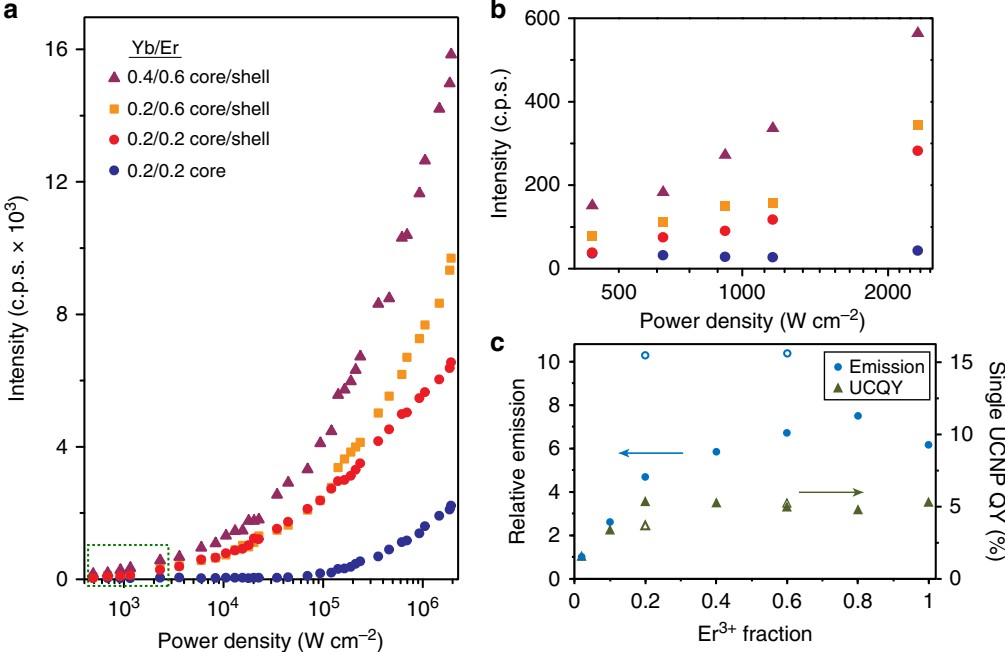

**Fig. 3** UCNP emission as a function of laser intensity. **a** Single 8-nm UCNP emission (496–745 nm) as a function of 980 nm laser excitation density. **b** Emission at low intensities from highlighted area (green dash) in **a**. **c** Emission (circles) and QYs (triangles) of single core/shell UCNPs (solid symbols) and aUCNPs (open symbols) relative to 20% $Yb^{3+}$ 2% $Er^{3+}$ core/shell UCNPs[13]. Values are based on averages of between 50 and 300 single UCNPs excited at a power density of $2 \times 10^6$ W cm$^{-2}$. Relative QYs are calculated as in Methods

these single UCNP experiments suggest it is not relief from concentration quenching that drives increases in aUCNP emission, but rather concentration enhancement in effective absorption cross-section due to the close proximity and larger ground state populations of of $Ln^{3+}$ able to absorb incident photons (Fig. 5). Concentration enhancement presents a second

role for the emitter ion $Er^{3+}$, in increasing absorbance directly, or indirectly, by desaturating $Yb^{3+}$ and enabling it to absorb more photons per unit time. This unexpected role of $Er^{3+}$ in absorption also answers the question of why aUCNPs are brighter in cases where neither number of absorbing $Yb^{3+}$ ions nor the QYs change significantly.

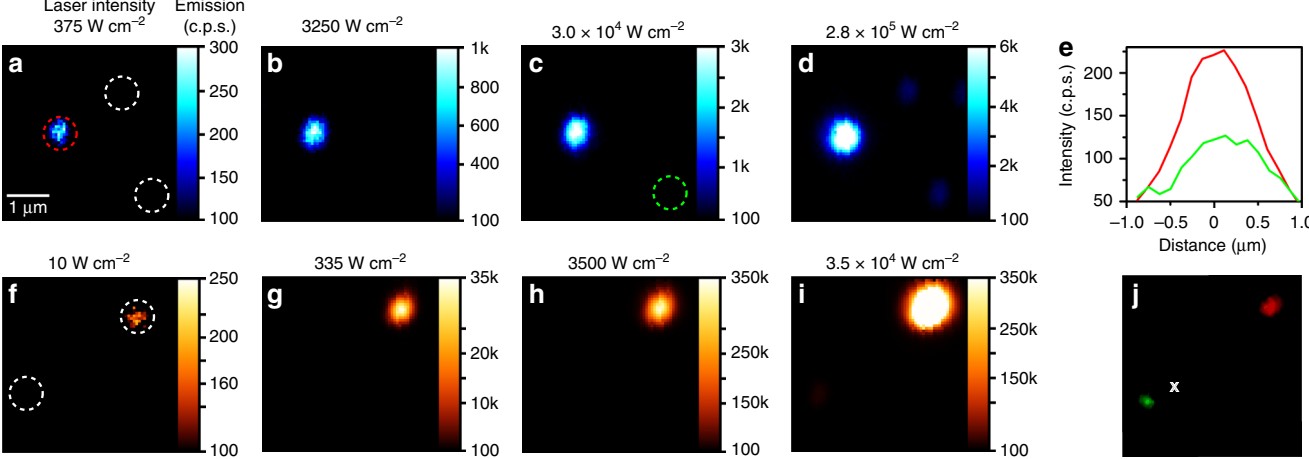

**Fig. 4** Multiphoton imaging at low irradiance. **a–d** Images of single core/shell 8-nm NaEr$_{0.2}$Yb$_{0.8}$F$_4$ aUCNPs (red dashed circle) and 8-nm NaYF$_4$: 20% Yb$^{3+}$, 20% Er$^{3+}$ UCNPs (white dashed circles), with emission intensity scale in c.p.s. next to each image. Scale bar is 1 μm. **e** Linecuts of single NaEr$_{0.2}$Yb$_{0.8}$F$_4$ core/shell aUCNP at 375 W cm$^{-2}$ (red circle) and NaYF$_4$: 20% Yb$^{3+}$, 20% Er$^{3+}$ UCNP at 30,000 W cm$^{-2}$ (green circle). **f–i** Images in 0.5-μm polystyrene beads loaded with ensembles of UCNPs (compositions as in **a–d**). **j** Brightfield image of the two beads in **f–i**, with 980-nm laser focused at the X

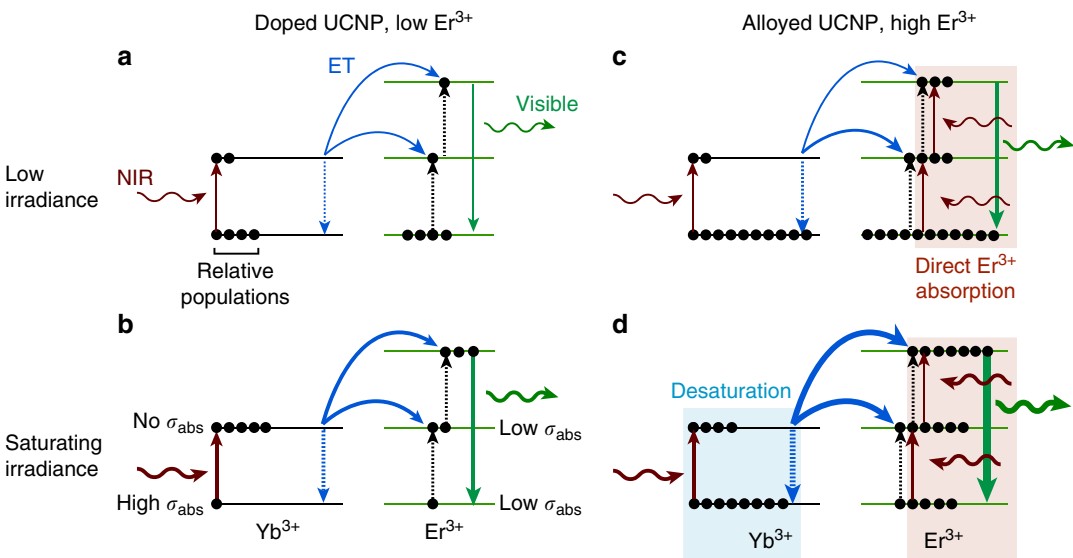

**Fig. 5** Mechanisms of enhanced aUCNP absorption. Simplified energy diagrams of doped (**a**, **b**) and alloyed (**c**, **d**) UCNPs at low (**a**, **c**) and high (**b**, **d**) irradiances. Energy transfer (ET) is denoted by blue arrows, absorption/emission by colored arrows, and arrow thickness reflects ET rates. At higher Er$^{3+}$ content, Er$^{3+}$ enhances aUCNP absorbance at 980 nm ($\sigma_{abs}$) indirectly through desaturation of the Yb$^{3+}$ excited state above saturating laser intensities, and through direct absorption at both low and high laser intensities. Mechanistic pathways determined by kinetic simulations as in Supplementary Methods

To examine whether the higher Ln$^{3+}$ content of aUCNPs lead to the faster Er$^{3+}$-Yb$^{3+}$ ET rates that underlie concentration enhancement, we examined lifetime decays of doped and alloyed UCNPs as a function of excitation intensity (Supplementary Fig. 9). At saturating intensities ($I_s > 3000$ W cm$^{-2}$), weighted lifetimes of core/shell aUCNP green and red emission decrease with increasing Er$^{3+}$ fraction, from ~400 μs at 100 W cm$^{-2}$ to ~10 μs at 10$^6$ W cm$^{-2}$ (Supplementary Fig. 8b, c), without associated quenching manifested as decreases in QY or brightness (Fig. 3c). Above $I_s$, models of time-resolved luminescence[18,34] have found strong correlations between decays of emitting Er$^{3+}$ levels and excited Yb$^{3+}$ ions, which act as reservoirs that rapidly repopulate Er$^{3+}$ emitting levels via ET. Shortened aUCNP lifetimes are therefore consistent with the onset of rapid Yb$^{3+}$

desaturation pathways mediated by close Er$^{3+}$–Yb$^{3+}$ pairs. In contrast, at sub-saturation intensities, UCNPs show similar decays regardless of Er$^{3+}$ content (Supplementary Fig. 9), again suggesting that lifetimes of emitting Er$^{3+}$ ions are influenced more by the kinetics of the Yb$^{3+}$ excited state than by other relaxation processes[23,25]. This unusual combination of brighter emission with shorter radiative lifetimes may be useful for fast-scanning techniques such as confocal imaging, where the long lifetimes of doped UCNPs can lead to blurring[35].

The sharp power dependence of aUCNP lifetimes suggests different mechanisms of emission enhancement above and below $I_s$ (Fig. 5). At higher powers that saturate Yb$^{3+}$ absorption, enhanced aUCNP emission depends on rapid ET to proximal Er$^{3+}$ ions, desaturating Yb$^{3+}$ excited states so that the Yb$^{3+}$ ions

are freed to absorb incident photons. Below $I_s$, UCNPs show similar lifetimes regardless of $Er^{3+}$ content. At these low excitation intensities, most $Yb^{3+}$ ions are already in their ground states, enhancement in aUCNPs is driven primarily by increased absorption due to the larger number of absorbing $Ln^{3+}$ ions per aUCNP. Small increases in total aUCNP $\sigma_{abs}$ at 980 nm (Supplementary Fig. 8a) are significant because of the quadratic dependence of emission on the photon absorption rate. For example, the calculated 980-nm $\sigma_{abs}$ of $NaYb_{0.4}Er_{0.6}F_4$ aUCNPs is twice that of UCNPs doped with 20 $Yb^{3+}$ and 20% $Er^{3+}$, which suggests a fourfold higher emission and which aligns well with the experimentally measured 4.2-fold enhancement at 490 W $cm^{-2}$ (Fig. 3b). Combined with calculations and lifetime decays, quantitative single UCNP measurements across a broad range of $Ln^{3+}$ content and excitation intensities have allowed us to distinguish the critical mechanisms of aUCNP emission enhancement that dominate at different imaging conditions.

**Deep-tissue imaging of 12-nm aUCNPs.** To determine how aUCNPs can be imaged in biological systems at low laser intensities, we transferred 8-nm $NaEr_{0.6}Yb_{0.4}F_4$ aUCNPs with 2-nm shells (12 nm total diameter) to water via polymer encapsulation[36] and injected them into mammary fat pads 3–4 mm beneath the skin of 5-week-old mice (Fig. 6 and Methods). Images of green emission acquired with 980-nm excitation at 0.1 W $cm^{-2}$ all show SBR $\geq$25, with signal decreasing from 2 to 6 h, likely owing to a slow extravasation of aUCNPs from the mammary glands into draining lymph nodes. By comparison, the maximum permissible exposure for 980 nm continuous-wave lasers to human skin is 0.73 W $cm^{-2}$ [28,37], and previous deep-tissue experiments with doped UCNPs have typically required far larger doped UCNPs to achieve similar SBR values at these low irradiances[28,38]. These experiments demonstrate that protein-sized aUCNPs can be locally injected and imaged without notable toxicity, and demonstrate a robust aUCNP signal from deep tissue, even with visible emission.

## Discussion

Increasingly complex multi-shell and multi-$Ln^{3+}$ designs for UCNPs have been successfully deployed for super-resolution imaging, lasing, and sensitized emission[21,39–41]. Here we find a

new role for $Er^{3+}$ in enhancing absorption, leading to simple core/shell designs with just $Yb^{3+}$ and $Er^{3+}$ as optically active ions that can yield superior nanoparticles over the entire range of useful UCNP imaging intensities. The addition of inert shells radically changes the brightest compositions from doped UCNPs to alloyed, dispensing with the need for a host matrix. These newly optimized compositions show >300-fold emission enhancements compared to doped UCNPs, and aUCNPs just 12 nm in diameter can be imaged deep in tissue at excitation intensities well below those known to cause physiological damage[4,5,28,37]. While we have focused on aUCNPs similar in size to antibodies to maintain biocompatibility, increasing the sizes of these same aUCNPs would further decrease the minimum required intensities, as brightness scales with the number of $Ln^{3+}$, which increase as $d^3$. Larger aUCNPs may limit certain applications in confined or crowded systems, but open up other applications where extremely low intensities of low-energy NIR light are required to minimize phototoxicity, such as with stem cells, embryos, or human tissue.

## Methods

**Synthesis of alloyed $\beta$-$NaEr_xYb_yF_4$ nanocrystals.** aUCNP nanocrystals were synthesized using a previously described method[19] for Ln-doped $\beta$-$NaYF_4$ with some modifications. For $NaEr_{0.2}Yb_{0.8}F_4$ aUCNPs: to a dry 50-mL round bottom flask, $YbCl_3$ hydrate (0.32 mmol, 127 mg) and $ErCl_3$ hydrate (0.08 mmol, 22 mg) were added, followed by oleic acid (3.25 g, 10.4 mmol) and 1-octadecene (ODE, 4 mL). The flask was stirred, placed under vacuum, and heated to 110 °C for 1 h, causing the solution to become clear. The flask was cooled and filled with $N_2$, and sodium oleate (1.25 mmol, 381 mg), $NH_4F$ (2 mmol, 74 mg), oleylamine (1.25 mL, 0.38 mmol), and ODE (1.75 mL) were added. The flask was again placed under vacuum and stirred at room temperature for 20 min, and then flushed three times with $N_2$. The reaction was heated to 315 °C, stirred for 45 min under $N_2$, and then cooled rapidly by a strong stream of air to the outside of the flask following removal of the heating mantle. When the reaction had cooled to 75 °C, ethanol (20 mL) and acetone (20 mL) were added to precipitate the nanocrystals. The reaction was transferred to a centrifuge tube and centrifuged at $3000 \times g$ for 5 min to precipitate the nanocrystals completely. The supernatant was discarded and the white solid (80 mg) was resuspended in minimal hexanes (5 mL) to break up the pellet. The nanocrystals were then precipitated again with the addition of ethanol (45 mL) and centrifuged at $3000 \times g$ for 5 min. The nanocrystals were resuspended and stored in 10 mL of hexanes with 0.2% (v v$^{-1}$) oleic acid to give a 10 µM dispersion.

Similar procedures were used for all $NaEr_xYb_yF_4$, $NaErF_4$, and doped $NaYF_4$ cores, except that no oleylamine was required for nanocrystals with <40% Yb. All nanocrystals showed pure $\beta$ phase by powder XRD and TEM.

**Synthesis of core/shell aUCNPs.** Epitaxial $NaY_{0.8}Gd_{0.2}F_4$ shells were grown on $NaEr_xYb_yF_4$, $NaErF_4$, and doped $NaYF_4$ cores using a method modified from that of Li[29]. The added $Gd^{3+}$ was found to be necessary for pure $\beta$-phase shell growth[13]. Precursors were prepared by heating $YCl_3$ (0.40 mmol, 78 mg) and $GdCl_3$ (0.10 mmol, 26 mg) to 110 °C in oleic acid (2 mL) and ODE (3 mL) and stirred for 15 min under vacuum. The flask was filled with $N_2$ and heated to 160 °C for 30 min to allow the $GdCl_3$ to dissolve, which was followed by another 15 min at 110 °C under vacuum, to give a 0.10 M solution of 80:20 Y/Gd oleate (Y/Gd-OA). In a separate flask, a Na/F precursor was prepared by dissolving sodium trifluoroacetate (1.20 mmol, 163 mg) in oleic acid (3 mL) and applying vacuum at room temperature for 20 min, giving a 0.40 M Na-TFA-OA solution.

For 4-nm shells grown on 8-nm cores: a stock dispersion of $NaEr_xYb_yF_4$ cores (27 µmol in hexane) was added to 4 mL of oleic acid and 6 mL of ODE. Hexane was removed by applying vacuum at 70 °C for 30 min, $N_2$ was introduced, and the flask heated to 280 °C. Injection cycles of Y/Gd-OA and Na-TFA-OA precursor volumes[29] (Supplementary Table 1) were followed 5 min after the reaction reached to 280 °C. Every cycle began with the Y/Gd-OA solution, followed by the Na-TFA-OA solution 15 min later to form single 0.5-nm unit cell layer. After the last injection, the reaction was maintained at 280 °C for 30 min to allow for complete shell growth. Then, the reaction was cooled rapidly, and 15 mL of ethanol and 15 mL of acetone were added when the reaction reached 75 °C. Nanoparticles were precipitated, cleaned, and stored as described for the UCNP cores.

**Nanocrystal characterization.** For XRD, 1 mL of a stock solution of the nanoparticles in hexane was precipitated with the addition of 2 mL of EtOH. The nanoparticle slurry was spotted onto a glass coverslip or silicon wafer multiple times until an opaque white film formed, and the sample was allowed to air dry completely. XRD patterns were obtained on a Bruker AXS D8 Discover GADDS X-ray diffractometer system with Co K$\alpha$ radiation ($\lambda = 1.78897$ Å) from 2$\theta$ of 15 to 65°.

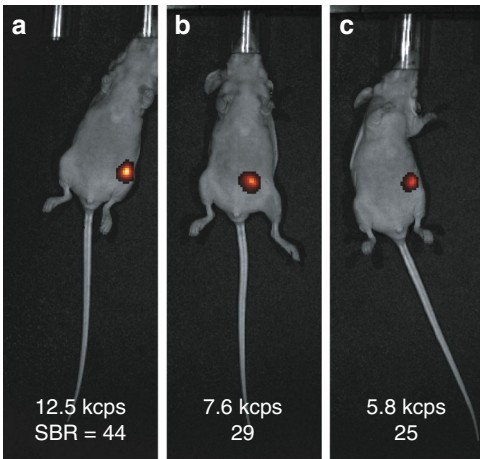

**Fig. 6** Deep-tissue imaging of aUCNPs at low excitation intensity. Imaging of 12-nm core/shell $NaEr_{0.6}Yb_{0.4}F_4$ (8-nm core wth 2-nm shell) aUCNPs injected into mammary fat pads 3–4 mm below the skin at **a** 2 h, **b** 4 h, and **c** 6 h after injection. Laser intensity is 0.1 W $cm^{-2}$ focused at or near the injection site, and emission is from the $Er^{3+}$ $^4S_{3/2}$ band (530–550 nm). SBR: signal-to-background ratio

Nanocrystal size was determined by dynamic light scattering measurements on a Malvern Zetasizer. Samples were prepared from hexane stocks by dilution with hexane to ~50 nM. The diameters of the nanoparticles in each sample were determined based on the fitting by volume according to instrument software.

For EM, UCNPs were precipitated from hexane with EtOH, washed with EtOH, and resuspended in hexane to 10 nM; 7 μL was dropped onto ultrathin carbon film/holey carbon grid, 400 mesh copper (Ted Pella). Images of the nanoparticles were obtained using a Zeiss Gemini Ultra-55 analytical scanning electron microscope. Dark-field images were collected in transmission (STEM) mode with 30 kV beam energy. TEM images were also obtained using a JEOL 2100-F 200 kV field-emission analytical transmission electron microscope. For single-particle determination, samples deposited on silicon nitride windows (Ted Pella) were used and imaged in STEM mode at 30 kV.

**Single UCNP imaging**. For emission spectra and power series, UCNPs were diluted in hexane to ~1 fM before dropcasting onto No.1 glass coverslip or silicon nitride grid (Ted Pella). Laser scanning confocal imaging was performed in ambient conditions using a lab-built pinhole confocal[18] (Supplementary Fig. 10) with 980-nm continuous-wave laser (Thorlabs TCLDM9, 300 mW diode). Because the diffraction-limited beam spot is larger than individual nanoparticle size, emission single particles was confirmed on SiN TEM-grid samples by correlation with subsequent SEM imaging on a Zeiss Ultra-55, operating in transmission mode. Single-particle determination was also confirmed by intensity histograms compiled from 50–300 individual UCNPs for each composition (Supplementary Fig. 11). From the confocal laser scanned images, the Gaussian distribution peaks were adopted as the single-particle upconversion emission intensity. Samples with ambiguous distributions or ones not correlating with SEM images were discarded. Excitation power series were created by rotating a series of neutral density filters in the path of the laser beam while monitoring power simultaneously.

Excitation power density (PD) was calculated as PD = $P/S$, where $P$ is the laser power on the sample and $S$ is the laser spot area. Laser powers at the sample were measured in-line by an NIST-traceable power meter (Thorlabs) with S120C sensor, covering 400–1100 nm and 50 nW–50 mW with 1 nW resolution. To calculate power densities (W cm$^{-2}$), the beam spot area was determined by fitting the UCNP emission profile to a Gaussian distribution, where the diameter is the full-width at half-max in the form FWHM = $2\sqrt{2\ln 2}\,\sigma$ of 580 nm, giving a laser spot area of $2.64 \times 10^{-9}$ cm$^2$.

**Absorption cross-sections and QYs**. Representative Er$^{3+}$ absorption cross-sections ($\sigma_{abs}$) at 980 nm were calculated using Judd–Ofelt theory in conjunction with experimental absorption measurements;[33] the Yb$^{3+}$ $\sigma_{abs}$ value was determined from literature values as in Supplementary Methods. Whole-UCNP $\sigma_{abs}$ values were determined by calculating number of Yb$^{3+}$ and Er$^{3+}$ ions per UCNP according to average TEM diameters (Supplementary Fig. 2c), Ln$^{3+}$ fractions, and hexagonal phase NaLnF$_4$ dimensions. Single UCNP QYs were determined relative to the standard value (QY = 0.49%)[13] of core/shell 8-nm 20% Yb$^{3+}$ 2% Er$^{3+}$ UCNPs measured at 10 W cm$^{-2}$, which was converted to a high-intensity value using calculated power dependence curve in Supplementary Fig. 8b. Single UCNP emissions from Fig. 2a were used, and all compositions were taken to be at 75% saturation based on kinetic simulations described in Supplementary Methods, Supplementary Tables 1 and 2, and Supplementary Fig. 12.

**Ensemble UCNP imaging**. To image aUCNP ensembles in beads, as described previously[13], UCNPs were loaded into 0.5 μm polystyrene beads (Aldrich) by swelling 0.5 mg of beads in 250 μL of a 5% (v v$^{-1}$) CHCl$_3$ solution in $n$-BuOH. aUCNPs (2 mg in 15 μL of hexane) were added to the bead suspension and briefly vortexed. After stagnant incubation at 25 °C for 4 h, the beads were washed two times with EtOH, centrifuged at 3000 × $g$ for 4 min, and stored in EtOH. Based on single-bead and single UCNP intensities, we estimate there are 70 UCNPs and 200 aUCNPs per bead in Fig. 4.

**Aqueous passivation of core/shell aUCNPs**. Hydrophobic 8-nm NaEr$_{0.6}$Yb$_{0.4}$F$_4$ aUCNPs with 2-nm NaY$_{0.8}$Gd$_{0.2}$F$_4$ shells were dispersed in hexane with 0.2% (v v$^{-1}$) oleic acid to 5 μM. For aqueous dispersions[36], 6 mg of poly(maleic anhydride-$alt$-1-octadecene) amphiphilic copolymer ($M_W$ 20–25k, Aldrich) was dissolved to 17 μM in 0.5 mL of acetone and 15 mL of CHCl$_3$. aUCNPs (0.5 nmol) in 100 μL of hexane were added with stirring, and the solvents were removed under a gentle stream of N$_2$ overnight. aUCNP/polymer residue was then resuspended in a solution of methoxy-PEG$_8$-amine (Thermo Fisher, 10 μmol) in 10 mL of 100 mM NaHCO$_3$ buffer, pH 8.2, with 1% (v v$^{-1}$) dimethyl sulfoxide. This suspension was sonicated for 60 min, heated in an 80 °C water bath for 60 min, slowly cooled to room temperature, and then sonicated for 30 min. Excess polymer was removed by extensive spin dialysis (Amicon, 100 kDa MWCO), washing with 7 × 15 mL of 100 mM HEPES (4-(2-hydroxyethyl)-1-piperazineethanesulfonic acid), pH 7.4. The retentate was concentrated to 680 μL and filtered through a 0.2-μm filter into a sterile glass vial. Final concentration (200 nM) was determined using an emission vs. concentration curve measured for the parent hydrophobic aUCNPs.

**Live animal imaging**. Animal experiments were conducted according to protocols approved by the UCSF Animal Care and Use Committee. Nude/nude homozygous, female, 5-week-old mice (Taconic Farms) were anesthetized and injected with 25 μL of 200 nM 12-nm core/shell NaEr$_{0.6}$Yb$_{0.4}$F$_4$ (8-nm core with 2-nm shell) dispersions into mammary fat pads 3–4 mm below the skin. Mice were imaged with an IVIS Spectrum In Vivo Imaging System (Perkin Elmer) equipped with a 4.5 mW 980-nm continuous-wave laser (Thorlabs) and 780-nm longpass filter (Chroma). The beam was focused to 0.1 W cm$^{-2}$ and emission was collected from 530–550 nm in the green Er$^{3+}$ band using 2.5-s integration times at 2, 4, and 6 h after injection. Dark counts were measured in the absence of laser excitation, and background was measured in uninjected areas with laser excitation. SBR values were calculated as the ratio of emission to background with dark counts subtracted from each.

**Statistical analysis**. Where indicated, results are presented as mean ± one standard deviation.

**Data availability**. All relative data are available from the corresponding authors upon request.

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

## Acknowledgements

We thank V. Mann and C. Ajo-Franklin for helpful discussions. B.T. and Y.T. were supported by fellowships from the Chinese Scholarship Council, and A.T. was supported by the Weizmann Institute of Science - National Postdoctoral Award Program for Advancing Women in Science. Work at the Molecular Foundry was supported by the Director, Office of Science, Office of Basic Energy Sciences, Division of Materials Sciences and Engineering, of the U.S. Department of Energy under Contract No. DE-AC02-05CH11231.

## Author contributions

B.E.C., P.J.S., and E.M.C. conceived the project and supervised the research. B.T., N.A.T., and C.A.T. synthesized the nanocrystals. B.T., A.F-B., A.T., Y.T., N.J.B., and E.S.B. conducted the optical experiments; M.V.P.A. and B.T. carried out the EM and other characterization; N.A.T., H.N., and M.A. conducted the animal imaging. E.M.C. carried out the calculations and modeling. B.T. and B.E.C. prepared the figures; B.T., E.M.C., P.J.S., and B.E.C. analyzed the results and wrote the manuscript. All authors participated in discussion and editing of the manuscript.
