## [Peer Review File · Nature Communications]

Reviewers' comments:

Reviewer #2 (Remarks to the Author):

Summary of key results, Additional Comments on 3rd Review: The authors have improved the manuscript and focused on reporting the properties of small (20nm), "alloyed" UCNPs. This reviewer appreciates the new figure and additional discussion on the mechanisms for the brighter UCNPs at low excitation intensity. The major contribution of the paper is the new composition(s) of the lanthanide doped (alloyed) UCNPs reported, which allow imaging 20nm UCNPs at lower excitation intensities than previous reports.

Originality and significance, Additional Comments on 3rd Review: Having read over the paper a number of times, I am surprised that the authors, while having tempered the discussion regarding potential, but undemonstrated, applications to bio-imaging, continue to compare CW excitation intensities used to excite sequential multi-photon energy transfer upconversion to the peak intensities used to excite instantaneous multi-photon processes. The author's loose usage of the term "fluence" throughout the paper, including the title, where intensity and fluence (energy density) are used interchangeably throughout the paper, may be misleading. Usually we consider fluence when we are using pulsed excitation and are considering the initial population of a time-resolved experiment. I am not sure why we are using the term at all for the CW excitation case. It seems the authors are assuming that phototoxicity is only a function of time-averaged energy.

For the average powers and intensities cited in the paper (300 W/cm^2), typical fluence for a high rep rate femtosecond laser (typically used in multiphoton imaging) would be only 0.15 J/cm^2 . The time averaged power would be 4×10^4 times higher than what is quoted for single UCNP sensitivity (e.g. 30mW compared to 750nW), but the issue for bioimaging is what is causing phototoxicity, is it absorption or a non-linear effect? This would need to be assessed before making any prediction about which method of excitation would be preferable, and either method may actually be superior in a particular application.

In bioimaging, I believe low average power is desirable in instances wherein absorption is leading to phototoxicity, what is the level of absorption at 980nm should be determined before making a claim that the particles would be superior in (all?) applications. If non-linear effects are important, we would need to identify these as being the primary cause of photo-toxicity before asserting that upconversion nanoparticles are a superior solution. Again, I am not diminishing the authors work, but the comparisons being made in the paper still concern me.

Suggested improvements, Additional Comments on 3rd Review: Again, I want to compliment the authors on their achievements in developing novel nano-materials. As noted above, I believe the discussion has improved. However, I think there are still some vestiges of the prior arguments relating to the suitability of these UCNPs for bio-imaging and comparisons to other methods and applications wherein these materials may have valuable but unproven applications, that are weakening the paper.

I would suggest strongly scrutinizing the term fluence everywhere it's used, and omitting statements like lines 26-30 "and almost 9 orders of magnitude lower than for 2-photon imaging of single

fluorophores¹⁵. These fluences are lower even than those required for a number of one-photon single molecule techniques¹⁶, multiphoton imaging of ensembles of fluorophores^{17,18}, and certain super-resolution techniques^{19,20}." Any discussions of phototoxicity should be supported by evidence.

The new discussion about the mechanisms for the improvements in brightness of the UCNPs are interesting, but could be developed more thoroughly and should be supported by evidence.

This reviewer is appreciative of the novel properties of the materials developed, and very familiar with many of the proposed applications. A caution about being clear and making solid claims may help this paper to move through review more easily and avoid placing erroneous or speculative claims into archival literature.

In its present form, I do not think the paper is appropriately placed in Nature Communications. The novelty, and the emphasis, in my opinion, should be on the photophysics of these materials, but needs further development. Future potential applications seem promising, but these need to be demonstrated to show their relevance.

Reviewer #3 (Remarks to the Author):

The so called "alloyed UCNP" is not new. Synthesis and characterization of alloyed Yb-Er, Yb-Tm, and Yb-Ho systems all have been reported, although the term "aUCNP" was not used. The authors cannot just ignore the previous developments. In the current study, the authors tuned the compositions of the alloyed nanoparticles and used new methodology to characterize the nanoparticles. The use of low fluences to see individual alloyed UCNPs is impressive. But the 3000-fold reduction in critical fluences is exaggerated because of unfair comparisons. High excitation power used in STEM is for stimulated-emission-depletion, not for photon excitation of the nanoparticles. For comparison of excitation powers, the benchmark was not properly selected. It seems that the authors are not aware of the existing 'alloyed' UCNPs.

Overall, a real application of the nanoparticles in bioimaging is expected to prove the significant advancement over previous studies including the authors own work, and to appeal to the broad readership of Nature Communication.

Other comments:

1: Fig. 3c is not cited or discussed in the manuscript. The authors should explain what is this figure intended for and what are the implications of this figure.

2. Fig. 4 is not discussed in the manuscript either. Only the last paragraph of the method section mentioned Fig. 4 very briefly.

We appreciate the efforts the reviewers, and we have fully addressed all questions and concerns below. Our responses are in **blue**, changes to the manuscript are in **bold**, and the changes to the manuscript are “quoted”.

Reviewer #3 (Remarks to the Author):

The so called “alloyed UCNP” is not new. Synthesis and characterization of alloyed Yb-Er, Yb-Tm, and Yb-Ho systems all have been reported, although the term “aUCNP” was not used. The authors cannot just ignore the previous developments. In the current study, the authors tuned the compositions of the alloyed nanoparticles and used new methodology to characterize the nanoparticles.

We appreciate the reviewer’s point and are certainly aware of these studies. **As described below, we have increased the discussion and number of citations for this general class of UCNP.** We in no way meant to suggest to have been the first to report all-Ln NaLnF₄ UCNPs.

We have introduced the descriptor “alloyed” to describe concisely and accurately UCNPs where all heavy metals are Ln³⁺ ions participating in upconversion, while avoiding oxymoronic phrases such as “100% doped”. Our goal has been to be clear and concise, and to distinguish these UCNPs from the vast majority that have been reported. We are not aware of any prior use of *alloyed* for UCNPs or any systematic and quantitative studies comparing alloyed with doped UCNPs.

The key advance of this paper is being able to image both single aUCNPs and ensembles at laser intensities far lower than previously reported for comparably sized doped UCNPs. We have attempted to emphasize this from the first 3 words of the title, and with details in the abstract and throughout the manuscript. To make this comparison, we have carried out a systematic study using strictly quantitative single particle comparisons over five orders of magnitude excitation intensity, which show that alloyed UCNPs are brighter than doped UCNPs at all laser powers. We can therefore image them at >300-fold lower power densities than previously reported for similarly sized doped UCNPs. To develop a mechanistic understanding of these, we include power-dependent lifetimes, quantum yield, calculations, and kinetic simulations, all of which allow us to develop a new mechanism (“concentration enhancement”) that differs sharply from previous UCNP mechanisms.

We fully agree with the reviewer about the importance of accurately representing prior work, and want to continue to make every effort capture related studies to the best of our knowledge. **We had cited previous NaLnF₄ systems with high Ln³⁺ content in this sentence (with multiple references) in the introduction, and based on the reviewer’s suggestion, we have added new references for each of the compositions below:**

“For UCNPs with high Ln³⁺ content, this has been attributed to suppression of “concentration quenching”, an observation that encompasses a number of known as well as unexplored energetic pathways that reduce the quantum yield of upconverted emission²⁷⁻³².”

We have also added a sentence about synthesis of sub-10 nm beta phase aUCNPs, which we believe has not previously been reported, and include an additional 4 references to all-Ln UCNPs as pointed out by the reviewer:

“We synthesized a series of 8-nm diameter β-phase NaYF₄ cores²⁵, and overcoated them with NaYF₄ shells using a layer-by-layer protocol³³ (see Methods). Several NaLnF₄ alloys of heavy lanthanides (e.g., Yb³⁺-Er³⁺, Yb³⁺-Tm³⁺, Yb³⁺-Ho³⁺, as well as NaErF₄) have been reported^{28,32,34-36}, and because heavy Ln³⁺ favor larger nanoparticles^{11,25,36}, these compositions have been unable to access the brighter β-phase at sizes under 25 nm.”

While we are careful to avoid making claims of primacy in the text, we emphasize here the novel findings of our manuscript:

- As noted above, our most efficient compositions can now be imaged at >300 -fold lower power densities than previously reported for similarly sized doped UCNPs. Specific discussion of this comparison is given below.

- This is the first single-particle imaging of any all-Ln NaLnF₄ UCNPs, and this gives us strictly quantitative comparisons of these and related compositions over 5 orders of magnitude power density. This is a level of quantitative comparison not possible previously in cuvettes or single-power measurements.

In Figure 2a, we show single UCNP imaging for 12 compositions we believe have never been imaged at the single particle level. In the main text and SI, we show single particle power series over 5 orders of magnitude laser intensity for 9 compositions we believe never characterized so thoroughly. We know of no other experimental method able to make strict quantitative comparisons between UCNP compositions over such a large range of excitation powers, and these span the useful microscopy powers from live animal imaging all the way up to single-molecule and superresolution techniques.

- While we stress the imaging results as the key findings of this study, we believe this is the first reported synthesis of sub-10 nm beta-phase all-Yb/Er NaLnF₄. As the reviewer is likely aware, beta phase nanoparticles are 2-3 orders of magnitude brighter than comparable alpha phase nanoparticles, and smaller nanocrystals are essential for many imaging applications. There was a significant synthetic challenge in small beta UCNPs with high Yb/Er content that we have overcome in this paper.

- We use single-particle quantum yield measurements to show that suppression of “concentration quenching”, often cited as a mechanism for low emission in high-Ln UCNPs, cannot be used to fully explain the increases in emission we measure. We introduce the concept of “concentration enhancement” based on increases in absorbance cross section at higher Ln content, and this is supported by quantum yields, power-dependent lifetimes, and simulations. For Yb/Er systems, this is also, we believe, the first time Er has been described as playing a significant role in UCNP absorbance, rather than as simply an emitter.

- We have added an important application: deep-tissue imaging of aUCNPs in live mice, showing high contrast at a laser intensity of just 0.1 W cm⁻² to image small nanoparticles several millimeters deep.

The use of low fluences to see individual alloyed UCNPs is impressive. But the 3000-fold reduction in critical fluences is exaggerated because of unfair comparisons. High excitation power used in STEM is for stimulated-emission-depletion, not for photon excitation of the nanoparticles. For comparison of excitation powers, the benchmark was not properly selected.

The >300 -fold reduction in minimum laser intensity (please note, not 3000-fold) we make is a comparison from *data within this paper* (and in fact, within the same figure: Figure 3a), so this is an exact comparison. This benchmark was identical to a power series for 8-nm beta-NaYF₄ with 20% Yb and 20% Er from *Nat Nano* 9, 300 (2014), which we reproduced for this study.

We do appreciate that comparisons between imaging modalities are inherently inexact. **Our goal was to provide some context for our power densities in terms of familiar imaging techniques, but given the reviewers' objections, we have removed all comparisons to other imaging techniques.** We will rely on (1) the internal >300 -fold reduction in intensities needed to image single nanoparticles, and (2) the

added deep tissue imaging data (described below) which gives strong emission 3-4 mm deep at just 0.1 W cm^{-2} , well below the ANSI standard of 0.73 W cm^{-2} 980-nm CW for human skin.

It seems that the authors are not aware of the existing ‘alloyed’ UCNP. Overall, a real application of the nanoparticles in bioimaging is expected to prove the significant advancement over previous studies including the authors own work, and to appeal to the broad readership of Nature Communication.

Based on the reviewers’ suggestion of an application, we have added deep tissue, live animal aUCNP imaging, showing that we can image ensembles of core/shell $\text{NaEr}_{0.4}\text{Yb}_{0.6}\text{F}_4$ aUCNPs, 12 nm in total diameter, 3-4 mm deep at murine mammary fat pads. Notably, an excitation power density of 0.1 W cm^{-2} gives us signal:background >25 . We have added this as Figure 5:

Figure 5: Deep tissue imaging of aUCNPs at low excitation intensity. Imaging of 12-nm core/shell $\text{NaEr}_{0.6}\text{Yb}_{0.4}\text{F}_4$ (8-nm core with 2-nm shell) aUCNPs injected into mammary fat pads 3-4 mm below the skin at (a) 2 h, (b) 4 h, and (c) 6 h after injection. Laser intensity is 0.1 W cm^{-2} focused at or near the injection site, and emission is from the $\text{Er}^{3+} \text{ } ^4\text{S}_{3/2}$ band (530 – 550 nm). SBR, signal to background ratio.

We have also added this to the abstract

"Core/shell aUCNPs 12 nm in total diameter can be imaged with strong contrast (signal:background >25) through several millimeters of tissue in live mice using a laser intensity of 0.1 W cm^{-2} ."

And to the Discussion:

“To determine how aUCNPs can be imaged in biological systems at low laser intensities, we transferred 8-nm $\text{NaEr}_{0.6}\text{Yb}_{0.4}\text{F}_4$ aUCNPs with 2-nm shells (*i.e.*, 12 nm total diameter) to water *via* polymer encapsulation³⁹ and injected them into mammary fat pads 3-4 mm beneath the skin of 5-week-old mice (Fig. 5 and Methods). Images of green emission acquired with 980-nm excitation at 0.1 W cm^{-2} all show $\text{SBR} \geq 25$, with signal decreasing from 2 to 6 h, likely owing to a slow extravasation of aUCNPs from the mammary glands into draining lymph nodes. By comparison, the maximum permissible exposure for 980 nm continuous wave lasers to human skin is 0.73 W cm^{-2} (refs. 32, 40), and previous deep-tissue experiments with doped UCNP have typically required far larger doped UCNP to achieve similar SBR values at these low irradiances^{32,41}. These experiments demonstrate that protein-sized aUCNPs can be locally

injected and imaged without notable toxicity, and demonstrate a robust aUCNP signal from deep tissue, even with visible emission.”

And we have added detailed procedures to the Methods section:

“*Aqueous passivation of core/shell aUCNPs.* Hydrophobic 8-nm $\text{NaEr}_{0.6}\text{Yb}_{0.4}\text{F}_4$ aUCNPs with 2-nm $\text{NaY}_{0.8}\text{Gd}_{0.2}\text{F}_4$ shells were dispersed in hexane with 0.2% (v/v) oleic acid to 5 μM . For aqueous dispersions³⁹, 6 mg of poly(maleic anhydride-*alt*-1-octadecene) amphiphilic copolymer (M_w 20-25k, Aldrich) was dissolved to 17 μM in 0.5 mL of acetone and 15 mL of CHCl_3 . aUCNPs (0.5 nmol) in 100 μL of hexane were added with stirring, and the solvents were removed under a gentle stream of N_2 overnight. aUCNP/polymer residue was then resuspended in a solution of methoxy-PEG₈-amine (ThermoFischer, 10 μmol) in 10 mL of 100 mM NaHCO_3 buffer, pH 8.2, with 1% (v/v) DMSO. This suspension was sonicated for 60 min, heated in an 80 °C water bath for 60 min, slowly cooled to room temperature, and then sonicated for 30 min. Excess polymer was removed by extensive spin dialysis (Amicon, 100 kDa MWCO), washing with 7 × 15 mL of 100 mM HEPES, pH 7.4. The retentate was concentrated to 680 μL and filtered through a 0.2- μm filter into a sterile glass vial. Final concentration (200 nM) was determined using an emission *versus* concentration curve measured for the parent hydrophobic aUCNPs.

Animal Imaging. Animal experiments were conducted according to protocols approved by the UCSF Animal Care and Use Committee. Nude/nude homozygous, female, 5-week-old mice (Taconic Farms) were anesthetized and injected with 25 μL of 200 nM 12-nm core/shell $\text{NaEr}_{0.6}\text{Yb}_{0.4}\text{F}_4$ (8-nm core with 2-nm shell) dispersions into mammary fat pads 3-4 mm below the skin. Mice were imaged with an IVIS Spectrum In Vivo Imaging System (PerkinElmer) equipped with 4.5 mW 980-nm continuous wave laser (Thorlabs) and 780-nm longpass filter (Chroma). The beam was focused to 0.1 W cm^{-2} and emission was collected from 530 – 550 nm in the green Er^{3+} band using 2.5-s integration times at 2, 4, and 6 h after injection. Dark counts were measured in the absence of laser excitation, and background was measured in uninjected areas with laser excitation. SBR values were calculated as the ratio of emission to background with dark counts subtracted from each.”

We have also added 3 authors who carried out these experiments: Hossein Najafiaghdam, Nicole A. Torquato, and Mekhail Anwar.

Other comments:

1: Fig. 3c is not cited or discussed in the manuscript. The authors should explain what is this figure intended for and what are the implications of this figure.

Thank you for bringing this to our attention. **We have replaced Fig 3c with single-UCNP quantum yields, which are a key part of the mechanistic discussion, and were previously part of Fig ED8:**

Figure 3: Characterization of single UCNP emission

a, Single 8-nm UCNP emission (496 – 745 nm) as a function of 980 nm laser excitation density. **b**, Emission at low fluences from highlighted area (green dash) in **a**. **c**, Emission and QYs of single core/shell UCNPs (solid circles) and aUCNPs (open circles) relative to 20% Yb³⁺ 2% Er³⁺ core/shell UCNPs¹³. Values are based on averages of between 50 and 300 single UCNPs excited at a power density of $2 \times 10^6 \text{ W cm}^{-2}$. Relative QYs are calculated as in Methods.

Please note that power-dependent lifetimes formerly included as Fig. 3c are included as Fig ED9 and now discussed at length in the main text:

" To examine whether the higher Ln³⁺ content of aUCNPs lead to the faster Er³⁺-Yb³⁺ ET rates that underlie concentration enhancement, we examined lifetime decays of doped and alloyed UCNPs as a function of excitation intensity (Extended Data Fig. 9). At saturating intensities ($I_s > 3000 \text{ W cm}^{-2}$), weighted lifetimes of core/shell aUCNP green and red emission decrease with increasing Er³⁺ fraction, from $\sim 400 \mu\text{s}$ at 100 W cm^{-2} to $\sim 10 \mu\text{s}$ at 10^6 W cm^{-2} (Extended Data Fig. 8b, c), without associated quenching manifested as decreases in quantum yield or brightness (Fig. 3c). Above I_s , models of time-resolved luminescence^{13,38} have found strong correlations between decays of emitting Er³⁺ levels and excited Yb³⁺ ions, which act as reservoirs that rapidly repopulate Er³⁺ emitting levels via ET. Shortened aUCNP lifetimes are therefore consistent with the onset of rapid Yb³⁺ desaturation pathways mediated by close Er³⁺-Yb³⁺ pairs. In contrast, at sub-saturation intensities, UCNPs show similar decays regardless of Er³⁺ content (Extended Data Fig. 9), again suggesting that lifetimes of emitting Er³⁺ ions are influenced more by the kinetics of the Yb³⁺ excited state than by other relaxation processes²⁸. This unusual combination of brighter emission with shorter radiative lifetimes may be useful for fast-scanning techniques such as confocal imaging, where the long lifetimes of doped UCNPs can lead to blurring⁴³.

The sharp power dependence of aUCNP lifetimes suggests different mechanisms of emission enhancement above and below I_s . At higher powers that saturate Yb³⁺ absorption, enhanced aUCNP emission depends on rapid ET to proximal Er³⁺ ions, desaturating Yb³⁺ excited states so that the Yb³⁺ ions are freed to absorb incident photons. Below I_s , UCNPs show similar lifetimes regardless of Er³⁺ content. At these low excitation intensities, most Yb³⁺ ions are already in their ground states, enhancement in aUCNPs is driven primarily by increased absorption due to the larger number of absorbing Ln³⁺ ions per aUCNP. Small

increases in total aUCNP σ_{abs} at 980 nm (Extended Data Fig. 8a) are significant because of the quadratic dependence of emission on the photon absorption rate. For example, the calculated 980-nm σ_{abs} of NaYb_{0.4}Er_{0.6}F₄ aUCNPs is twice that of UCNPs doped with 20 Yb³⁺ and 20% Er³⁺, which suggests a 4-fold higher emission and which aligns well with the experimentally-measured 4.2-fold enhancement at 490 W cm⁻² (Fig. 3b). Combined with calculations and lifetime decays, quantitative single UCNPs measurements across a broad range of Ln³⁺ content and excitation intensities have allowed us to distinguish the critical mechanisms of aUCNP emission enhancement that dominate at different imaging conditions."

2. Fig. 4 is not discussed in the manuscript either. Only the last paragraph of the method section mentioned Fig. 4 very briefly.

Figure 4 is included in the Discussion (underline added here):

"We observe that increasing the fraction of Yb³⁺ (Extended Data Fig. 5) enables single core/shell NaEr_{0.2}Yb_{0.8}F₄ aUCNPs to be imaged as low as 290 W cm⁻² (Figs. 4a-e and Extended Data Fig. 7), >300-fold lower than the best previous sub-20-nm compositions^{13,14}. For confocal imaging of ensembles of nanoparticles, fluences as low as 4 W cm⁻² can be used to image these same aUCNPs loaded into polystyrene beads (Figs. 4f-j and Extended Data Fig. 7)."

As Fig. 4 is a key finding, its data are also highlighted in the Abstract:

"Here, we develop protein-sized, alloyed UCNPs (aUCNPs) that can be imaged at the single particle level at laser intensities below 300 W cm⁻² (or, 750 nW at the sample), >300-fold lower than needed for comparably-sized doped UCNPs....Core/shell aUCNPs are brighter than comparably sized doped UCNPs at all laser intensities tested, over 5 orders of magnitude."

Reviewer #2 (Remarks to the Author):

Summary of key results, Additional Comments on 3rd Review: The authors have improved the manuscript and focused on reporting the properties of small (20nm), "alloyed" UCNPs. This reviewer appreciates the new figure and additional discussion on the mechanisms for the brighter UCNPs at low excitation intensity. The major contribution of the paper is the new composition(s) of the lanthanide doped (alloyed) UCNPs reported, which allow imaging 20nm UCNPs at lower excitation intensities than previous reports.

We appreciate the reviewer's significant effort in evaluating our manuscript, and based on previous suggestions regarding potential bioimaging applications, **we have added in deep-tissue imaging in live mice using 12-nm aUCNPs (described in more detail below) as a demonstration of small aUCNPs. In the abstract, we have added:**

"Core/shell aUCNPs 12 nm in total diameter can be imaged with strong contrast (signal:background >25) through several millimeters of tissue in live mice using a laser intensity of 0.1 W cm⁻²."

Because small changes in size can have large effects, we have noted in the Abstract that these nanoparticles are 12 nm. Please note that there are no 20 nm UCNPs in the paper and for clarity we have removed any mention of "sub-20 nm". (If they were 20 nm (16 nm core + 2 nm shell), they would be 8x brighter based on volumetric scaling.) All of the UCNPs in this study are 8 nm cores with either 2- or 4-nm shells.

The emphasis on SBR is based on the reviewer's previous comments, and we appreciate the guidance.

Originality and significance, Additional Comments on 3rd Review: Having read over the paper a number of times, I am surprised that the authors, while having tempered the discussion regarding potential, but undemonstrated, applications to bio-imaging, continue to compare CW excitation intensities used to excite sequential multi-photon energy transfer upconversion to the peak intensities used to excite instantaneous multi-photon processes. The author's loose usage of the term "fluence" throughout the paper, including the title, where intensity and fluence (energy density) are used interchangeably throughout the paper, may be misleading. Usually we consider fluence when we are using pulsed excitation and are considering the initial population of a time-resolved experiment. I am not sure why we are using the term at all for the CW excitation case. It seems the authors are assuming that phototoxicity is only a function of time-averaged energy.

We agree on the need to be precise in our language and have changed *fluence* to *irradiance* or *intensity* throughout. Thank you for pointing out this distinction.

For the average powers and intensities cited in the paper (300 W/cm^2), typical fluence for a high rep rate femtosecond laser (typically used in multiphoton imaging) would be only 0.15 J/cm^2 . The time averaged power would be 4×10^4 times higher than what is quoted for single UCNP sensitivity (e.g. 30 mW compared to 750 nW), but the issue for bioimaging is what is causing phototoxicity, is it absorption or a non-linear effect? This would need to be assessed before making any prediction about which method of excitation would be preferable, and either method may actually be superior in a particular application.

In bioimaging, I believe low average power is desirable in instances wherein absorption is leading to phototoxicity, what is the level of absorption at 980 nm should be determined before making a claim that the particles would be superior in (all?) applications. If non-linear effects are important, we would need to identify these as being the primary cause of photo-toxicity before asserting that upconversion nanoparticles are a superior solution. Again, I am not diminishing the authors work, but the comparisons being made in the paper still concern me.

Our goal was to provide some context for our power densities in terms of familiar imaging techniques, but given the reviewers' objections, **we have removed comparisons to power densities of other imaging techniques.**

To make more meaningful comparisons on phototoxicity, we now cite in the common standard (e.g., OSHA, FDA) American National Standard for Safe Use of Lasers value for maximum permitted dose of 980-nm CW for human skin, 0.73 W cm^{-2} (see below). By comparison, we are able to image 12-nm aUCNPs 3-4 nm deep in mice at just 0.1 W cm^{-2} , with signal:background ratios >25 . This is a direct comparison to an accepted, if very conservative, standard for phototoxicity using 980-nm CW lasers.

"To determine how aUCNPs can be imaged in biological systems at low laser intensities, we transferred 8-nm $\text{NaEr}_{0.6}\text{Yb}_{0.4}\text{F}_4$ aUCNPs with 2-nm shells (*i.e.*, 12 nm total diameter) to water *via* polymer encapsulation³⁹ and injected them into mammary fat pads $3\text{-}4 \text{ mm}$ beneath the skin of 5-week-old mice (Fig. 5 and Methods). Images of green emission acquired with 980-nm excitation at 0.1 W cm^{-2} all show $\text{SBR} \geq 25$, with signal decreasing from 2 to 6 h, likely owing to a slow extravasation of aUCNPs from the mammary glands into draining lymph nodes. By comparison, the maximum permissible exposure for 980 nm continuous wave lasers to human skin is 0.73 W cm^{-2} (refs. 32, 40), and previous deep-tissue experiments with doped UCNPs have typically required far larger doped UCNPs to achieve similar SBR values at these low irradiances^{32,41}. These experiments demonstrate that protein-sized aUCNPs can be locally

injected and imaged without notable toxicity, and demonstrate a robust aUCNP signal from deep tissue, even with visible emission.”

We have added this as Figure 5:

Figure 5: Deep tissue imaging of aUCNPs at low excitation intensity. Imaging of 12-nm core/shell $\text{NaEr}_{0.6}\text{Yb}_{0.4}\text{F}_4$ (8-nm core with 2-nm shell) aUCNPs injected into mammary fat pads 3-4 mm below the skin at (a) 2 h, (b) 4 h, and (c) 6 h after injection. Laser intensity is 0.1 W cm^{-2} focused at or near the injection site, and emission is from the $\text{Er}^{3+} {}^4\text{S}_{3/2}$ band (530 – 550 nm). SBR, signal to background ratio.

And we have added detailed procedures to the Methods section:

“*Aqueous passivation of core/shell aUCNPs.* Hydrophobic 8-nm $\text{NaEr}_{0.6}\text{Yb}_{0.4}\text{F}_4$ aUCNPs with 2-nm $\text{NaY}_{0.8}\text{Gd}_{0.2}\text{F}_4$ shells were dispersed in hexane with 0.2% (v/v) oleic acid to $5 \mu\text{M}$. For aqueous dispersions³⁹, 6 mg of poly(maleic anhydride-*alt*-1-octadecene) amphiphilic copolymer (M_w 20-25k, Aldrich) was dissolved to $17 \mu\text{M}$ in 0.5 mL of acetone and 15 mL of CHCl_3 . aUCNPs (0.5 nmol) in 100 μL of hexane were added with stirring, and the solvents were removed under a gentle stream of N_2 overnight. aUCNP/polymer residue was then resuspended in a solution of methoxy-PEG₈-amine (ThermoFischer, 10 μmol) in 10 mL of 100 mM NaHCO_3 buffer, pH 8.2, with 1% (v/v) DMSO. This suspension was sonicated for 60 min, heated in an 80 °C water bath for 60 min, slowly cooled to room temperature, and then sonicated for 30 min. Excess polymer was removed by extensive spin dialysis (Amicon, 100 kDa MWCO), washing with $7 \times 15 \text{ mL}$ of 100 mM HEPES, pH 7.4. The retentate was concentrated to 680 μL and filtered through a 0.2- μm filter into a sterile glass vial. Final concentration (200 nM) was determined using an emission *versus* concentration curve measured for the parent hydrophobic aUCNPs.

Animal Imaging. Animal experiments were conducted according to protocols approved by the UCSF Animal Care and Use Committee. Nude/nude homozygous, female, 5-week-old mice (Taconic Farms) were anesthetized and injected with 25 μL of 200 nM 12-nm core/shell $\text{NaEr}_{0.6}\text{Yb}_{0.4}\text{F}_4$ (8-nm core with 2-nm shell) dispersions into mammary fat pads 3-4 mm below the skin. Mice were imaged with an IVIS Spectrum In Vivo Imaging System (PerkinElmer) equipped with 4.5 mW 980-nm continuous wave laser (Thorlabs) and 780-nm longpass filter (Chroma). The beam was focused to 0.1 W cm^{-2} and emission was collected from 530 – 550 nm in the green Er^{3+} band using 2.5-s integration times at 2, 4, and 6 h after injection. Dark counts

were measured in the absence of laser excitation, and background was measured in uninjected areas with laser excitation. SBR values were calculated as the ratio of emission to background with dark counts subtracted from each.”

We have also added 3 authors who carried out these experiments: Hossein Najafiaghdam, Nicole A. Torquato, and Mekhail Anwar.

We would never suggest any one probe is superior in all applications, and we agree with the reviewer that there may be different mechanisms of phototoxicity for pulsed vs. continuous excitation. While this is far from our fields of expertise, we note that the ANSI and other laser safety standards list maximum permitted exposures, by wavelength, in W cm^{-2} for CW lasers. Regardless of excitation modality, reducing phototoxicity in bioimaging applications has always been a major goal, and phototoxicity in imaging has been studied exhaustively by others for at least half a century (e.g., *Nature* 201, 316 (1964)).

We have included 3 citations with NIR absorption profiles, which vary by type of tissue. We have also added a reference to a laser trapping experiment in which red blood cells have been optically trapped with CW NIR lasers in living mice without observed tissue damage (Nat Comm 4:1768 (2013)).

For specifics in addressing the reviewer’s questions, we quote from two of these references here. These studies find that NIR excitation is less toxic than comparable visible excitation, for both CW or pulsed experiments. These comparisons are made in the absence of probes.

“Near infrared (NIR) excitation is more benign than these higher energy wavelengths^{5,25}, and non-linear multiphoton techniques that use NIR excitation have been widely adopted^{1,2,6,10,26}.”

5 Squirrell, J. M., Wokosin, D. L., White, J. G. & Bavister, B. D. Long-term two-photon fluorescence imaging of mammalian embryos without compromising viability. *Nat Biotechnol* **17**, 763-767, (1999).

This paper compares the development of embryos by LSCM visible excitation to pulsed 2P excitation, showing that even 1000-fold higher NIR excitation was benign compared to visible:

“We have found that the development of hamster embryos was dramatically impaired by imaging with LSCM (flux density = $9 \times 10^3 \text{ W/cm}^2$; 8 μs dwell time)..... In striking contrast, embryo viability is maintained when embryos are imaged using the same microscope system with a 1,047 nm ultrashort pulsed laser (flux density = $6 \times 10^6 \text{ W/cm}^2$; 8 μs dwell time).”

These experiments are carried out in the absence of added luminescent probes, so the measured phototoxicity is only a function of the incident laser.

In addition, we have cited a UCNP tracking paper:

19 Nam, S. H. *et al.* Long-Term Real-Time Tracking of Lanthanide Ion Doped Upconverting Nanoparticles in Living Cells. *Angew Chem Int Edit* **50**, 6093-6097 (2011).

“It is well known, however, that the optical absorption cross sections of cellular components are dramatically reduced in the NIR range,^{25,26} and thus the chances of subsequent photoinduced damage upon NIR excitation are minimized. In fact, we found that 980 nm excitation is noninvasive to the living cells as expected. For example, even after illuminating a single HeLa cell for 6 h without interruption, the cell remained viable as manifested by the observation that trypan blue treatment did not stain the nucleus (Figure 1 b, upper panel). In contrast, when a different cell was illuminated with a visible laser (532 nm)

for comparison, it was dead in 1 h even with a power of 1 W cm^{-2} (Figure 1 b, lower panel), which is two orders of magnitude lower than that of 980 nm excitation.”

Because of the reviewer’s emphasis on this topic, we have added this sentence with 2 additional supporting citations about the steep wavelength-dependent absorbance of tissue:

“Both scatter and absorption by cellular components are much lower for NIR light than for visible light^{6,21,22}, and this steep wavelength dependence has been shown in direct comparisons to reduce photodamage using NIR-based techniques^{5,9,19,23}.”

Suggested improvements, Additional Comments on 3rd Review: Again, I want to compliment the authors on their achievements in developing novel nano-materials. As noted above, I believe the discussion has improved. However, I think there are still some vestiges of the prior arguments relating to the suitability of these UCNP for bio-imaging and comparisons to other methods and applications wherein these materials may have valuable but unproven applications, that are weakening the paper.

We appreciate the reviewer's gracious comments about the novelty of our nanoparticles. As noted above, we have heeded the reviewer's advice, removing direct comparisons to other techniques and adding in deep-tissue imaging showing strong signal from 12-nm aUCNPs at very low laser intensities.

I would suggest strongly scrutinizing the term fluence everywhere it’s used, and omitting statements like lines 26-30 “and almost 9 orders of magnitude lower than for 2-photon imaging of single fluorophores¹⁵. These fluences are lower even than those required for a number of one-photon single molecule techniques¹⁶, multiphoton imaging of ensembles of fluorophores^{17,18}, and certain super-resolution techniques^{19,20}.”

In addition to changing *fluence* to *irradiance* or *intensity* throughout, we have eliminated the offending sentence and all related language.

Any discussions of phototoxicity should be supported by evidence.

As described above, we have added ANSI values as a clear standard for 980 nm CW phototoxicity. We also point the reviewer to the cited references for experiments demonstrating major phototoxicity differences between visible and NIR excitation (with either pulsed or CW lasers), as well as the references describing differences in tissue absorbance between visible and NIR light. Our phototoxicity discussion falls squarely within the scope of these points; other topics (e.g., toxicity mechanisms) are covered by references to excellent studies in the literature.

The new discussion about the mechanisms for the improvements in brightness of the UCNP are interesting, but could be developed more thoroughly and should be supported by evidence.

We have added single UCNP quantum yields to Fig 3c to emphasize some of the evidence of our mechanistic conclusions. We have also added a discussion of intensity-dependent lifetime data and how this supports observed increases in brightness.

Figure 3: Characterization of single UCNP emission

a, Single 8-nm UCNP emission (496 – 745 nm) as a function of 980 nm laser excitation density. **b**, Emission at low intensities from highlighted area (green dash) in **a**. **c**, Emission and QYs of single core/shell UCNPs (solid circles) and aUCNPs (open circles) relative to 20% Yb³⁺ 2% Er³⁺ core/shell UCNPs¹³. Values are based on averages of between 50 and 300 single UCNPs excited at a power density of 2×10^6 W cm⁻². Relative QYs are calculated as in Methods.

Based on the reviewer's suggestion, we have added 2 paragraphs to the lifetime data (time-resolved emission for 8 compositions, each measured over 4 orders of magnitude) supporting our mechanism. :

" To examine whether the higher Ln³⁺ content of aUCNPs lead to the faster Er³⁺-Yb³⁺ ET rates that underlie concentration enhancement, we examined lifetime decays of doped and alloyed UCNPs as a function of excitation intensity (Extended Data Fig. 9). At saturating intensities ($I_s > 3000$ W cm⁻²), weighted lifetimes of core/shell aUCNP green and red emission decrease with increasing Er³⁺ fraction, from ~ 400 μ s at 100 W cm⁻² to ~ 10 μ s at 10^6 W cm⁻² (Extended Data Fig. 8b, c), without associated quenching manifested as decreases in quantum yield or brightness (Fig. 3c). Above I_s , models of time-resolved luminescence^{13,38} have found strong correlations between decays of emitting Er³⁺ levels and excited Yb³⁺ ions, which act as reservoirs that rapidly repopulate Er³⁺ emitting levels via ET. Shortened aUCNP lifetimes are therefore consistent with the onset of rapid Yb³⁺ desaturation pathways mediated by close Er³⁺-Yb³⁺ pairs. In contrast, at sub-saturation intensities, UCNPs show similar decays regardless of Er³⁺ content (Extended Data Fig. 9), again suggesting that lifetimes of emitting Er³⁺ ions are influenced more by the kinetics of the Yb³⁺ excited state than by other relaxation processes²⁸. This unusual combination of brighter emission with shorter radiative lifetimes may be useful for fast-scanning techniques such as confocal imaging, where the long lifetimes of doped UCNPs can lead to blurring⁴³.

The sharp power dependence of aUCNP lifetimes suggests different mechanisms of emission enhancement above and below I_s . At higher powers that saturate Yb³⁺ absorption, enhanced aUCNP emission depends on rapid ET to proximal Er³⁺ ions, desaturating Yb³⁺ excited states so that the Yb³⁺ ions are freed to absorb incident photons. Below I_s , UCNPs show similar lifetimes regardless of Er³⁺ content. At these low excitation intensities, most Yb³⁺ ions are already in their ground states, enhancement in aUCNPs is driven primarily by increased absorption due to the larger number of absorbing Ln³⁺ ions per aUCNP. Small

increases in total aUCNP σ_{abs} at 980 nm (Extended Data Fig. 8a) are significant because of the quadratic dependence of emission on the photon absorption rate. For example, the calculated 980-nm σ_{abs} of $\text{NaYb}_{0.4}\text{Er}_{0.6}\text{F}_4$ aUCNPs is twice that of UCNP doped with 20 Yb³⁺ and 20% Er³⁺, which suggests a 4-fold higher emission and which aligns well with the experimentally-measured 4.2-fold enhancement at 490 W cm⁻² (Fig. 3b). Combined with calculations and lifetime decays, quantitative single UCNP measurements across a broad range of Ln³⁺ content and excitation intensities have allowed us to distinguish the critical mechanisms of aUCNP emission enhancement that dominate at different imaging conditions."

This discussion derives from ED Fig 9:

Extended Data Figure 9. Power dependence of upconverted lifetimes. a Lifetime decay curves of UCNP at different excitation densities. Top row are 8-nm cores and bottom row are core/shell UCNP. **b** UCNP weighted lifetimes at different power densities. (*Left*) Weighted lifetime of green (²H_{11/2}, ⁴S_{3/2}) and (*right*) red (⁴F_{9/2}) Er³⁺ ion

emissions, obtained from exponential fitting of decay curves from **a**. Error bars are standard deviation propagated from uncertainties in exponential fittings and in some cases are smaller than data points.

We also have included, to support our mechanistic conclusions, in ED Fig 8, computational data about optical cross section and power-dependence of quantum yields:

Extended data Fig 8 Single UCNP absorbance cross sections and power-dependent quantum yields. **a** Calculated Er³⁺ and Yb³⁺ contributions to 8-nm UCNP absorbance cross sections (σ_{980}) as a function of Er³⁺ content. Yb³⁺ fraction is 0.2 except for NaErF₄. Values were calculated as in Methods. **b** QY variation of core/shell UCNPs with 20% Yb³⁺ as a function of laser power density. Values were calculated as in Methods and ref 1.

To support this, we also have included calculations of σ_{980} and simulations of QYs in the Methods and SI. Since this is fairly extensive, we have not cut-and-pasted it here. Because of the formatting of a previous draft, all of this evidence to support of our mechanistic conclusions is unfortunately spread out over 3 files.

This reviewer is appreciative of the novel properties of the materials developed, and very familiar with many of the proposed applications. A caution about being clear and making solid claims may help this paper to move through review more easily and avoid placing erroneous or speculative claims into archival literature.

In it's present form, I do not think the paper is appropriately placed in Nature Communications. The novelty, and the emphasis, in my opinion, should be on the photophysics of these materials, but needs further development. Future potential applications seem promising, but these need to be demonstrated to show their relevance.

We appreciate the advice of the reviewer and have taken heed. Specifically, we have demonstrated applications through high-SBR imaging of 12-nm aUCNPs in live mice at 0.1 W/cm² irradiance, well below established human phototoxicity standards. We have also worked to clarify our mechanistic conclusions, and include, in addition to quantitative single particle studies of almost 20 UCNPs, many over a 10,000-fold power density range: quantum yields of 9 different UCNPs, lifetimes of 8 UCNPs, each over 5 orders of magnitude laser intensity, calculations of absorbance cross sections, and state-of-

the-art modeling of saturation intensities and the power dependence of quantum yields. With the added 2 paragraphs of mechanistic discussion, we believe that this illustrates the validity of our claims.

We have added a figure and some small changes to the text to clear up our mechanistic discussion. These help to answer the question of how increasing Er^{3+} content can lead to brighter UCNPs without significant changes to quantum yields or to absorber Yb^{3+} content. While most discussions of UCNP brightness have previously focused on “concentration quenching” and other quantum yield-related mechanisms, we emphasize the new role for Er^{3+} in increasing absorbance through either *desaturation* or direct absorbance. This is shown in our new Fig 5:

Figure 5: Mechanisms of enhanced aUCNP absorbance

Simplified energy diagrams of doped (*left*) and alloyed UCNPs (*right*), at low (*top*) and high (*bottom*) irradiance. Energy transfer (ET) is denoted by blue arrows, absorbance/emission by colored arrows, and arrow thickness reflects ET rates. At higher Er^{3+} content, Er^{3+} enhances aUCNP absorbance indirectly through desaturation of the Yb^{3+} excited state above saturating laser intensities, and through direct absorbance at both low and high laser intensities.

We have also added sub-headers so the manuscript reads more clearly and added some small changes to the main text to introduce the new figure emphasize the novel role for Er^{3+} in absorbance:

“This unexpected role of Er^{3+} in absorbance also answers the question of why aUCNPs are brighter in cases where neither number of absorbing Yb^{3+} ions nor the quantum yields change significantly.”

and in the conclusion:

“Here we find a new role for Er^{3+} in enhancing absorbance, leading to that designs of simple core/shell nanoparticles designs with just Yb^{3+} and Er^{3+} as optically active ions that can yield superior nanoparticles over the entire range of useful UCNP laser imaging intensities.”

We think these changes bring clarity to the mechanistic insight supported by a large array of experimental and computational data. Thank you for encouraging us to make these changes to improve the manuscript.

REVIEWERS' COMMENTS:

Reviewer #2 (Remarks to the Author):

Summary of Key Results, Additional Comments on 4th Review:

The authors improved the manuscript by expanding the photo-physical characterization of their 12nm “alloyed” upconverting nanoparticles and adding a new figure describing the mechanism for improved brightness of these particles. More detailed photo-physical characterization, including power dependent luminescence decay were added. They further added a demonstration of sub-surface imaging of the particles in a live animal model at low excitation intensity (100mW/cm²).

Originality and Significance, Additional Comments on 4th Review:

I found the modified manuscript to be greatly improved, and appreciate the added material, especially figure 5, which illustrates the proposed mechanism for the high brightness of these materials. The authors should acknowledge the recent review on the challenges to obtaining high-brightness nano-materials using highly doped upconverting materials by Wen et. al. [7] and point out how their work fits within the strategies outlined in that review, which seem highly relevant. Extremely small and bright core-shell NaYbF₄Tm particles (7nm) were also reported recently by Shi et. al. [8], the authors should acknowledge the relevant work and point out any distinguishing characteristic of the reported upconverting nano-materials over seemingly similar Tm based materials. The additional demonstration of subsurface imaging of the UCNPs injected into a live animal model at notably low excitation intensity does add credibility to the claims of suitability of the materials for bio-imaging applications. The authors added some discussion on photo-toxicity, and rightly point out the known dependence on wavelength, which might be considered common knowledge, but including cell viability tests with UCNPs reported by Nam [9] certainly drives home the point. The authors cited a recent paper on mechanisms of photo-toxicity by Laissue [10], which relates, among other factors, the effect to generation of reactive oxygen species by not only the excitation light, but also the fluorophore emission, which UCNPs would also suffer from. However, the point of using long wavelength excitation is an inarguable improvement. I think it is appropriate to make this point, but to acknowledge that the exact mechanisms for photo-toxicity are complex and varied – most of which are to be determined for any given system.

Suggestions for improvements: Additional Comments on 4th Review:

I believe the manuscript is more focused and much improved. I believe the materials reported have some important features which will be of interest to workers in this field. Placing the work in the context of other efforts as suggested above would improve the presentation and highlight the significance of these findings. A key limitation for upconverting materials remains in the long lifetimes of the materials, which limit its applicability for dynamical and volumetric imaging – two very important features for bio-imaging. It would be appropriate in the concluding remarks to acknowledge this limitation and comment on the potential to overcome these limitations. Pointing out specific applications where upconverting nano-materials are an ideal candidate is also appropriate.

References:

- (7) Wen, S.; Zhou, J.; Zheng, K.; Bednarkiewicz, A.; Liu, X.; Jin, D. Advances in Highly Doped Upconversion Nanoparticles. Nat. Commun. 2018, 9, 2415.
- (8) Shi, R.; Ling, X.; Li, X.; Zhang, L.; Lu, M.; Xie, X.; Huang, L.; Huang, W. Tuning Hexagonal NaYbF₄ nanocrystals down to Sub-10 Nm for Enhanced Photon Upconversion. Nanoscale

2017, 9.

(9) Nam, S. H.; Bae, Y. M.; Park, Y. Il; Kim, J. H.; Kim, H. M.; Choi, J. S.; Lee, K. T.; Hyeon, T.; Suh, Y. D. Long-Term Real-Time Tracking of Lanthanide Ion Doped Upconverting Nanoparticles in Living Cells. *Angew. Chemie - Int. Ed.* 2011, 50, 6093–6097.

(10) Laissue, P. P.; Alghamdi, R. A.; Tomancak, P.; Reynaud, E. G.; Shroff, H. Assessing Phototoxicity in Live Fluorescence Imaging. *Nature Methods*, 2017, 14, 657–661.

Reviewer #3 (Remarks to the Author):

The authors have made major revisions to the manuscript with the addition of bioimaging data. The quality of the manuscript is now improved a lot. However, the previous works are still not properly acknowledged and thus the manuscript need to be further revised.

The statement “these compositions have been unable to access the brighter β -phase at sizes under 25 nm” is not true. Quite a few works on on the NaLnF₄ alloys of heavy lanthanides are available in literatures. And sub-10 nm β -phase NaYbF₄ UCNPs have been prepared and characterized last year (*Nanoscale*, 2017,9, 13739-13746). Therefore, the synthesis of nanoparticles for the current study should not be stressed as a highlight.

Response to reviewers for manuscript NCOMMS-18-07287B-Z

Reviewer #2 (Remarks to the Author):

Summary of Key Results, Additional Comments on 4th Review:

The authors improved the manuscript by expanding the photo-physical characterization of their 12nm “alloyed” upconverting nanoparticles and adding a new figure describing the mechanism for improved brightness of these particles. More detailed photo-physical characterization, including power dependent luminescence decay were added. They further added a demonstration of sub-surface imaging of the particles in a live animal model at low excitation intensity (100mW/cm²).

We appreciate the reviewer’s rereading and positive comments.

Originality and Significance, Additional Comments on 4th Review:

I found the modified manuscript to be greatly improved, and appreciate the added material, especially figure 5, which illustrates the proposed mechanism for the high brightness of these materials. The authors should acknowledge the recent review on the challenges to obtaining high-brightness nano-materials using highly doped upconverting materials by Wen et. al. [7] and point out how their work fits within the strategies outlined in that review, which seem highly relevant. Extremely small and bright core-shell NaYbF₄Tm particles (7nm) were also reported recently by Shi et. al. [8], the authors should acknowledge the relevant work and point out any distinguishing characteristic of the reported upconverting nano-materials over seemingly similar Tm based materials. The additional demonstration of subsurface imaging of the UCNPs injected into a live animal model at notably low excitation intensity does add credibility to the claims of suitability of the materials for bio-imaging applications. The authors added some discussion on photo-toxicity, and rightly point out the known dependence on wavelength, which might be considered common knowledge, but including cell viability tests with UCNPs reported by Nam [9] certainly drives home the point. The authors cited a recent paper on mechanisms of photo-toxicity by Laissue [10], which relates, among other factors, the effect to generation of reactive oxygen species by not only the excitation light, but also the fluorophore emission, which UCNPs would also suffer from. However, the point of using long wavelength excitation is an inarguable improvement. I think it is appropriate to make this point, but to acknowledge that the exact mechanisms for photo-toxicity are complex and varied – most of which are to be determined for any given system.

We wholeheartedly agree that more study of cellular phototoxicity, both as a function of irradiance and of imaging probe, would be useful in the development of novel imaging probes and techniques.

Suggestions for improvements: Additional Comments on 4th Review:

I believe the manuscript is more focused and much improved. I believe the materials reported

have some important features which will be of interest to workers in this field. Placing the work in the context of other efforts as suggested above would improve the presentation and highlight the significance of these findings. A key limitation for upconverting materials remains in the long lifetimes of the materials, which limit its applicability for dynamical and volumetric imaging – two very important features for bio-imaging. It would be appropriate in the concluding remarks to acknowledge this limitation and comment on the potential to overcome these limitations. Pointing out specific applications where upconverting nano-materials are an ideal candidate is also appropriate.

We have included this caveat on long lifetimes, with a reference to a van Veggel study, in discussing our lifetime data:

“This unusual combination of brighter emission with shorter radiative lifetimes may be useful for fast-scanning techniques such as confocal imaging, where the long lifetimes of doped UCNP_s can lead to blurring³⁵.”

References:

(7) Wen, S.; Zhou, J.; Zheng, K.; Bednarkiewicz, A.; Liu, X.; Jin, D. Advances in Highly Doped Upconversion Nanoparticles. *Nat. Commun.* 2018, 9, 2415.

(8) Shi, R.; Ling, X.; Li, X.; Zhang, L.; Lu, M.; Xie, X.; Huang, L.; Huang, W. Tuning Hexagonal NaYbF₄ nanocrystals down to Sub-10 Nm for Enhanced Photon Upconversion. *Nanoscale* 2017, 9.

(9) Nam, S. H.; Bae, Y. M.; Park, Y. Il; Kim, J. H.; Kim, H. M.; Choi, J. S.; Lee, K. T.; Hyeon, T.; Suh, Y. D. Long-Term Real-Time Tracking of Lanthanide Ion Doped Upconverting Nanoparticles in Living Cells. *Angew. Chemie - Int. Ed.* 2011, 50, 6093–6097.

(10) Laisue, P. P.; Alghamdi, R. A.; Tomancak, P.; Reynaud, E. G.; Shroff, H. Assessing Phototoxicity in Live Fluorescence Imaging. *Nature Methods*, 2017, 14, 657–661.

We agree these references are important and have add the Wen (ref 23) and Shi (ref 32) references. For Wen, this is an excellent new review, and we have added it here and other places:

“For UCNP_s with high Ln³⁺ content, this has been attributed to suppression of concentration quenching^{22,23}, an observation that encompasses a number of known as well as unexplored energetic pathways that reduce the quantum yield of upconverted emission^{22,24-28}.”

For Shi, we have added this to the list of NaLnF₄ nanocrystals and added a distinguishing remark, as suggested:

“Several NaLnF₄ alloys of heavy lanthanides (e.g., Yb³⁺-Er³⁺, Yb³⁺-Tm³⁺, Yb³⁺-Ho³⁺, as well as NaErF₄) have been reported^{22,25,28,30-32}, although these compositions have not been characterized by quantitative single particle imaging or systematically over a large range of power densities.”

Reviewer #3 (Remarks to the Author):

The authors have made major revisions to the manuscript with the addition of bioimaging data. The quality of the manuscript is now improved a lot. However, the previous works are still not properly acknowledged and thus the manuscript need to be further revised.

The statement “these compositions have been unable to access the brighter β -phase at sizes under 25 nm” is not true. Quite a few works on the NaLnF₄ alloys of heavy lanthanides are available in literatures. And sub-10 nm β -phase NaYbF₄ UCNPs have been prepared and characterized last year (Nanoscale, 2017,9, 13739-13746). Therefore, the synthesis of nanoparticles for the current study should not be stressed as a highlight.

We appreciate the reviewer’s rereading and positive comments. We have added the Shi reference (ref 32):

“Several NaLnF₄ alloys of heavy lanthanides (e.g., Yb³⁺-Er³⁺, Yb³⁺-Tm³⁺, Yb³⁺-Ho³⁺, as well as NaErF₄) have been reported^{22,25,28,30-31}, including sub-20 nm NaYbF₄:Tm core/shell nanoparticles³², although none of these compositions have not been characterized by quantitative single particle imaging or systematically over a large range of power densities.”